# Novel Zebrafish Mono-α2,8-sialyltransferase (ST8Sia VIII): An Evolutionary Perspective of α2,8-Sialylation

**DOI:** 10.3390/ijms20030622

**Published:** 2019-01-31

**Authors:** Lan-Yi Chang, Elin Teppa, Maxence Noel, Pierre-André Gilormini, Mathieu Decloquement, Cédric Lion, Christophe Biot, Anne-Marie Mir, Virginie Cogez, Philippe Delannoy, Kay Hooi Khoo, Daniel Petit, Yann Guérardel, Anne Harduin-Lepers

**Affiliations:** 1Université de Lille, CNRS, UMR 8576-UGSF-Unité de Glycobiologie Structurale et Fonctionnelle, F-59000 Lille, France; lanyi.chang@gmail.com (L.-Y.C.); maxence.noel@univ-lille.fr (M.N.); pierre-andre.gilormini@univ-lille.fr (P.-A.G.); mathieu.decloquement.etu@univ-lille.fr (M.D.); cedric.lion@univ-lille.fr (C.L.); christophe.biot@univ-lille.fr (C.B.); am.mir@wanadoo.fr (A.-M.M.); virginie.cogez@univ-lille.fr (V.C.); philippe.delannoy@univ-lille.fr (P.D.); yann.guerardel@univ-lille.fr (Y.G.); 2Institute of Biological Chemistry, Academia Sinica, Taipei 11529, Taiwan; kkhoo@gate.sinica.edu.tw; 3Sorbonne Université, Univ P6, CNRS, IBPS, Laboratoire de Biologie Computationnelle et Quantitative–UMR 7238, 4 Place Jussieu, 75005 Paris, France; elinteppa@gmail.com; 4Glycosylation et différenciation cellulaire, EA 7500, Laboratoire PEIRENE, Université de Limoges, 123 avenue Albert Thomas, 87060 Limoges CEDEX, France; daniel.petit@unilim.fr

**Keywords:** mono-α2,8-sialyltransferases, diSia motifs, evolution, ST8Sia, functional genomics

## Abstract

The mammalian mono-α2,8-sialyltransferase ST8Sia VI has been shown to catalyze the transfer of a unique sialic acid residues onto core 1 *O*-glycans leading to the formation of di-sialylated *O*-glycosylproteins and to a lesser extent to diSia motifs onto glycolipids like GD1a. Previous studies also reported the identification of an orthologue of the *ST8SIA6* gene in the zebrafish genome. Trying to get insights into the biosynthesis and function of the oligo-sialylated glycoproteins during zebrafish development, we cloned and studied this fish α2,8-sialyltransferase homologue. In situ hybridization experiments demonstrate that expression of this gene is always detectable during zebrafish development both in the central nervous system and in non-neuronal tissues. Intriguingly, using biochemical approaches and the newly developed in vitro MicroPlate Sialyltransferase Assay (MPSA), we found that the zebrafish recombinant enzyme does not synthetize diSia motifs on glycoproteins or glycolipids as the human homologue does. Using comparative genomics and molecular phylogeny approaches, we show in this work that the human ST8Sia VI orthologue has disappeared in the ray-finned fish and that the homologue described in fish correspond to a new subfamily of α2,8-sialyltransferase named ST8Sia VIII that was not maintained in Chondrichtyes and Sarcopterygii.

## 1. Introduction

Sialic acids are acidic monosaccharides mostly found at the outermost level of glycolipids and glycoproteins. Due to this terminal position and their nature, sialylated molecules are mediators for ligand-receptor and cell-cell interactions among other functions [1,2,3]. Furthermore, the function of a number of glycoconjugates at the cell surface depends on sialoglycan structures encountered (Neu5Ac, Neu5Gc, KDN), their modification (*O*-acetylation, *N*-acylation, *O*-methylation, *O*-sulfation), their glycosidic linkage (α2,3, α2,6, α2,8 and α2-8/9) and their degree of polymerization (DP) [4]. DiSia (DP = 2) structures are abundant on brain glycolipids of vertebrates and less abundant in glycoproteins [5], whereas oligoSia (2 < DP < 7) and polySia (DP >8) structures are prominent structural features of a restricted number of mammalian proteins conferring particular physico-chemical properties to these proteins and cell surfaces [6,7,8]. A few examples of di-sialylated proteins includes diSia structures linked to GalNAc residues on *O*-glycans from bovine chromogranins [9] or linked to *N*- and *O*-glycans in mouse B cells [10] and of human umbilical cord erythrocyte Band 3 [11] or linked to blood glycoproteins including von Willebrand factor core 1 *O*-glycan [12], immunoglobulins, MUC-1 [13], α_2_-macroglobulin and adipoQ from bovine serum [14,15].

In teleost fish, a broad variety of polySia chains have been described on the salmonid egg polysialoglycoprotein (PSGP) [16,17] with potential implication in osmoregulation during fertilization. In particular, zebrafish are characterized by a rich and diverse pattern of sialylation of glycoproteins and glycolipids that is temporally and spatially regulated. The *N*-glycome of the zebrafish embryo is characterized by the presence of both the canonical Siaα2-6Galβ1-4GlcNAc and the species-specific Galβ1-4(Siaα2-3)Galβ1-4(Fucα1-3)GlcNAc epitopes, whereas the *O*-glycome is dominated by the unusual Fucα1-3GalNAcβ1-4(Siaα2-3)Galβ1-3GalNAc and the conventional Siaα2-3/6Galβ1-4GlcNAcβ1-6(Siaα2-3Galβ1-3)GalNAc glycan structures [18,19]. Most of these sialylated structures are comprised of variable amounts of either Neu5Ac or Neu5Gc. Interestingly, fertilized eggs exhibit a stage-specific mucin-type *O*-glycan Fucα1-3GalNAcβ1-4(Siaα2-8Siaα2-3)Galβ1-3GalNAc with distinctive Neu5Ac/Neu5Gc sialylation pattern. Structural analysis clearly established that only Neu5Gc could be further elongated with another Sia residue generating Neu5Gcα2-8Neu5Gc and Neu5Acα2-8Neu5Gc di-sialylated motifs [18,20] suggesting high specificity of the enzymes involved in the synthesis of these zebrafish α2,8-sialylated epitopes. This unusual di-sialylated structure disappeared 24 h post fertilization (hpf) from the developing embryo. To summarize on the oligosialylation status of glycoconjugates during zebrafish development, it was shown that expression of glycoprotein-associated diSia motifs rapidly decreased following fertilization, whereas glycolipid-associated oligoSia (diSia and triSia) structures followed a reverse trend with an onset of expression at 24 hpf. Curiously, this pattern of oligosialylation did not correlate with the temporal expression of associated mono- oligo- and poly-α2,8-sialyltransferases and what we knew of their enzymatic activity suggesting other regulatory mechanisms [20]. In adult zebrafish tissues, oligosialylation was exclusively detected in brain gangliosides, in agreement with the high expression level of all α2,8-sialyltransferases identified in the zebrafish genome [21].

Biosynthesis and function of di-, oligo- and poly-sialylated glycoproteins during embryonic development of vertebrates and in their adult tissues is not well known due to the number of sialyltransferases involved that have not been well studied and enzymatically characterized, up to now. Indeed, one of the major challenges that glycobiologists have to face is to determine the enzymatic specificity of each newly identified vertebrate enzyme. Sialyltransferases belong to the GT CAZy family 29 [22], a subset of glycosyltransferases that was already present in the Last Common Ancestor of Eukaryotes (LECA) [23]. These biosynthetic enzymes are characterized by the presence in their protein sequence of four conserved motifs called sialylmotifs (L, S, III and VS) implicated in 3-D structure maintenance, substrate binding, and catalysis [24,25,26,27,28]. These sialylmotifs are also useful for in silico identification of homologous sialyltransferases in the genomic and transcriptomic databases and reconstruction of their evolutionary history [29,30,31,32]. Four families of sialyltransferases known as ST3Gal, ST6Gal, ST6GalNAc and ST8Sia are distinguished according to their substrate specificities and glycosidic linkage formed [31,33] and each family is characterized by family motifs likely involved in linkage specificity and acceptor monosaccharide recognition [29,34]. We have developed a general strategy to identify and assess phylogenetic distribution of sialyltransferases in more distantly related genomes and more importantly to infer sialyltransferase function [32]. Three of the four sialyltransferase families were studied in the context of the whole genome duplication (WGD) events that took place in the emerging vertebrates, e.g., two rounds of duplications at the base of vertebrates and a third round at the base of teleost fish [35,36,37,38]. This led to the identification in the fish genomes of a large set of novel β-galactoside α2,3/6-sialyltransferase-related sequences belonging to the ST3Gal and ST6Gal families [37,38,39] that could explain the recently described innovations in the fish sialome [18,19,20,21,40]. The human ST8Sia family is comprised of six subfamilies: ST8Sia I, ST8Sia V and ST8Sia VI are mono-α2,8-sialyltransferases involved in di-sialylation of glycoconjugates, while ST8Sia III are oligo-α2,8-sialyltransferases and ST8Sia II and ST8Sia IV are poly-α2,8-sialyltransferases implicated in the polysialylation of glycoproteins [29] and the six human orthologues could be identified in the zebrafish genome [31]. However, this last family appears to be much larger in teleost fish, since we noted the presence of α2,8-sialyltransferase-related sequences defining new subfamilies like the ST8Sia III-related (ST3Sia III-r) and the ST8Sia VII found in Cyprinidae and Salmonidae fish. This ST8Sia VII subfamily has arisen 552 million years ago (MYA) after the first whole genome duplication event (WGD-R1) and is also found in snakes and lizards, whereas the ST8Sia III-r subfamily is found in a few fish orders like perciformes, tetraodontiformes and beloniformes, and diverged from the ST8Sia III subfamily about 474 MYA following the second whole genome duplication event (WGD-R2) [35].

Following recent structural studies of α2,8-sialylation patterns conducted during zebrafish embryonic development [18,20] and in adult zebrafish tissues [21], we wondered whether the α2,8-linked sialic acids found on the zebrafish di-sialylated *O*-glycans could be transferred by the orthologue of the human ST8Sia VI identified in the zebrafish genome [31], since the human enzyme was reported to be responsible for the biosynthesis of di-sialylated *O*-glycosylproteins (Neu5Acα2,8Neu5Acα2,3Galβ1,4GalNAc-*O*-Ser). Interestingly, the human enzyme showed very low activity or no activity towards gangliosides or sialylated *N*-glycosylproteins [41], whereas the mouse ST8Sia VI showed slightly different specificity towards *O*-glycans and GM3 [42]. As a first step towards bringing a functional basis to the distribution of the various diSia motifs described in the zebrafish tissues, we analyzed the spatio-temporal profile of the zebrafish *st8sia6*-like gene during the zebrafish embryogenesis. In an attempt to define its biochemical activity, we expressed a recombinant enzyme and took advantage of chemoenzymatic glycan labeling strategies using unnatural CMP-activated sialic acid reporters [43,44] and of the newly developed MicroPlate Sialyltransferase Assay (MPSA) [45] to show that the enzyme was not active on sialylated fetuin. Towards understanding this unexpected data, we assessed the evolutionary relationships of fish mono-α2,8-sialyltransferases (i.e., ST8Sia I, ST8Sia V, ST8Sia VI and ST8Sia VII). Combining molecular phylogeny, sequence similarity network and synteny/paralogy analyses, we showed that the human ST8Sia VI orthologue disappeared in teleosts fish genomes, whereas another distinct mono-α2,8-sialyltransferase subfamily renamed ST8Sia VIII was present in Teleost fishes and had disappeared in Chondrichtyes (sharks) and Sarcopterygii (lobbed-finned fishes and tetrapods).

## 2. Results and Discussion

### 2.1. In Silico Identification and Sequence Analysis of Zebrafish ST8Sia Sequence

In an initial investigation to identify the zebrafish α2,8-sialyltransferases, the human ST8Sia VI nucleotide sequence (AJ621583, [41]) used as a query to screen genomic databases [31] led to the identification of a single zebrafish sequence (AJ715551) using the BLAST algorithm [46]. Exhaustive searches in the transcriptomic database [47] did not yield evidence for another ST8Sia VI-related sequence. The zebrafish gene sequence was located on the zebrafish chromosome 3, spanning approximately 16 kb. It was shown to contain seven exons producing a 1080 bp transcript, and to present a genomic organization comparable to the one described for the human *ST8SIA6* gene [29,31,35]. Translation of the open reading frame predicted a polypeptide of 360 amino acids containing the four sialylmotifs (L, S, III and VS) characteristic of all the sialyltransferases of the GT#29 CAZy family [23] and also the family-motifs “a” (NPSI) and “b” (GFWPF) (Figure 1) predicted for the α2,8-sialyltransferases [29,34,35]. Hydropathy analysis of this protein indicated the presence of a 19 amino acid hydrophobic sequence in the NH_2_-terminal region likely corresponding to the transmembrane domain. Five potential *N*-glycosylation sites were also predicted in the zebrafish ST8Sia VI-like sequence (Figure 1), which are not conserved in the human nor in the mouse ST8Sia VI sequences as observed in multiple sequence alignment (data not shown). Intriguingly, the zebrafish sequence showed a rather low degree of identity (37%) compared to its human orthologue (Table 1), whereas the other zebrafish ST8Sia proteins showed a higher percentage of identity ranging from 59% to 78% to their human counterparts (Table 2). Furthermore, this zebrafish sequence had comparable levels of similarity with the human ST8Sia V sequence. Altogether, these first sequence analyses indicated that the zebrafish ST8Sia VI-like protein was likely evolutionarily related to these mono-α2,8-sialyltransferases, although it may have undergone rapid evolution in the fish genome.

### 2.2. Spatio-Temporal Expression of the st8sia-Like Gene during Zebrafish Development

To determine the role of this *st8sia*-like gene in zebrafish, we next investigated the spatio-temporal expression of the zebrafish gene during embryonic development. Quantitative RT-PCR (QPCR) was used to analyze its expression in staged embryos. Our data shown in Figure 2A indicated an increased pattern of temporal expression along zebrafish embryo development, whereas diSia motifs on *O*-glycosylproteins decreased as previously mentioned [20]. We also examined the distribution of the *st8sia6*-like transcripts during zebrafish embryonic and larval development using whole mount in situ hybridization (ISH). A probe complementary to the *st8sia6*-like cDNA was designed, and ISH was performed on whole zebrafish embryos from different developmental stages. The hybridization sites were revealed by a chromogenic reaction with digoxigenin and the expression patterns were analyzed. The transcripts were not detected in the gastrula (i.e., 5 hpf) and were first detected at the early segmentation stage (i.e., 10 hpf; 1–4 somites stage), showing a general distribution of the mRNA (Figure 2B). During middle segmentation stage (i.e., 17 hpf; 10–13 somite stages), the *st8sia6*-like transcripts showed a general expression and were reinforced in the ventral portion of the spinal cord, in somites and in primordial pharyngeal arches. In the pharyngula stage, at 24 hpf, we detected general expression of the transcripts with higher staining in the myotomes, pharyngeal arches, heart and hypaxial muscles. At 36 hpf, the *st8sia6*-like gene expression was restricted to the ventricular part of the central nervous system (CNS), to pronephric ducts, to axial vasculature (caudal vein and posterior cardinal vein), to hypothalamus and dorsal part of the hindbrain. At hatching stage (48 hpf), expression of the *st8sia6*-like transcripts was detected in the dorsal region of rhombencephalon, in the optic tectum, in the pectoral fin, as well as in dorsal part of telencephalon and diencephalon. At larval stage (5 days), expression was detected in the dorsal rhombencephalon, in the ventricular part of the midbrain and around the epiphysis. This pattern of expression of the zebrafish *st8sia6*-like gene further suggested a role of this enzyme during zebrafish development, not only in the central nervous system, but also in non-neuronal biological systems and at the whole animal level. Previous studies have shown that the zebrafish mono-, oligo- and poly-α2,8-sialyltransferase genes exhibit distinct and overlapping pattern of expression in the developing central nervous system [48,49,50,51,52]. Indeed, a recent study highlighted the importance of the zebrafish *st8sia6*-like gene as a Dlx5-transcriptional target in the developing zebrafish olfactory/GnRH system that could confer anti-adhesive properties to neuronal surfaces [53]. Interestingly, the zebrafish *st8sia6*-like gene is also expressed in non-neuronal tissues like the pharyngeal arches and somites, similar to the previously described zebrafish *st8sia3* gene [48] indicating a potential role of this enzyme during muscle development.

### 2.3. Expression of a Recombinant and Soluble Protein—Enzymatic Characterization

To facilitate functional analyses, a soluble form of the zebrafish enzyme, cytoplasmic and transmembrane domains deleted, was constructed in the expression vector 3xFLAG-CMV9. The truncated cDNA lacking the first 33 amino acid residues of the *N*-terminus region (Δ33ST8Sia VI-like) was transiently transfected in mammalian cells. As expected, the recombinant FLAG-tagged protein could be detected in the culture medium and cell lysate of the transfected COS-7 cells by Western blot using BioM2 anti-FLAG monoclonal antibody. Although the theoretical molecular mass of the recombinant zebrafish enzyme was 42 kDa, we detected several protein isoforms ranging from 50 to 52 kDa, further suggesting the presence of post-translational modifications (Figure 3). Indeed, we showed that these bands corresponded to *N*-glycosylated isoforms since peptide N-glycosidase F (PNGase F) treatment induced a shift to the expected 42 kDa. However, it is interesting to note that the relative expression and secretion levels of the zebrafish Δ33ST8Sia VI-like were quite low, similar to the one described for the human Δ27ST8Sia VI [41,54]. It is now accepted that *in vivo*, most glycosyltransferases assemble to form homodimers or heterooligomeric complexes with other proteins and that these processes modulate their enzymatic activity [55,56]. Sequences flanking the transmembrane domain are often involved in these interactions and thus their removal or the absence of essential co-factors may explain the encountered difficulties in protein folding and secretion.

As a first attempt to characterize the zebrafish ST8Sia VI-like biochemical function, we next studied the enzymatic activity of the soluble enzyme produced in the cell culture medium of transiently transfected cells using in vitro enzymatic assays, fetuin and labeled CMP-[^14^C]Neu5Ac, as previously described for other sialyltransferases [44,57,58]. Native fetuin possesses three sialylated *O*-glycans and three sialylated *N*-glycans [59,60], and served as a good acceptor substrate to characterize the human ST8Sia VI enzymatic activity [41]. However, we only could detect extremely low levels of transfer of [^14^C] sialic acid residues onto the sialylated-glycans of fetuin, preventing further structural and DP analyses (data not shown). Failure to detect significant enzymatic activity prompted us to use expression vectors with the complete fish ST8Sia VI-like cDNA sequence FLAG-tagged or not, to transiently transfect COS-7 cells. Microsomal fractions containing the full length fish enzyme were used in enzymatic assays, but again, no significant enzymatic activity could be detected (data not shown). We therefore chose to apply the quick and sensitive MPSA developed recently to assess the sialyltransferase activity of human recombinant ST3Gal I and ST6Gal I onto glycoprotein acceptors [45]. In this assay, the acceptor glycoprotein is coated on the bottom of 96-well plate and the crude sialyltransferase activity is assessed using CMP-Sia*N*Al, a high-energy donor form of the unprotected alkyne-tagged sialic acid reporter Sia*N*Al that is readily used by these two human sialyltransferases [43,44] followed by covalent ligation of an azido-probe via a Cu(I) catalyzed azide-alkyne cycloaddition (CuAAC) [61,62]. Firstly, we optimized the MPSA for the human recombinant ST8Sia VI and defined the optimum reaction conditions (i.e., temperature, incubation time and proper substrates concentrations) for this ST8Sia enzyme. We carried out a time course assay and observed the time-dependent Sia*N*Al transfer activity of the human ST8Sia VI onto fetuin reaching a plateau at 4 h, whereas no Sia*N*Al transfer activity was detected for the zebrafish enzyme, or the mock control (Figure 4A). Indeed, we observed a significant Sia*N*Al transfer activity of the human ST8Sia VI onto native fetuin, bovine submaxillary gland mucin (BSM) and orosomucoid and almost no transfer activity onto asialofetuin and asialoorosomucoid (Figure 4B), as previously described using CMP-[^14^C]Neu5Ac [41]. These data showed that the MPSA approach could be used to determine mono-α2,8-sialyltransferase activities onto various glycoprotein acceptors. However, no significant Sia*N*Al transfer activity could be detected for the zebrafish enzyme, whatever the isoform of the enzyme used or the acceptor substrate used (i.e., mammalian glycoproteins) (Figure 4B) or GD1a ganglioside (data not shown). These data further suggested a loss of function of this enzyme or a low affinity for the used mammalian acceptor substrates and a quite different enzymatic activity of the zebrafish enzyme. Recent studies and the present data using the human ST8Sia VI have shown that vertebrate sialyltransferases tolerate large modifications at the C-5 position [44,63] and so a preference of the zebrafish enzyme for CMP-Neu5Gc over CMP-Neu5Ac is also less probable. There are only a handful of studies reporting enzymatic characterization of fish sialyltransferases, which all suggest lower activities of the fish enzymes compared to their mammalian orthologue. For instance, the zebrafish orthologue of the human ST8Sia IV, which is primarily involved in the polysialylation of the N-CAM *N*-glycans also showed very low levels of transfer activity from CMP-Neu5Ac onto murine N-CAM compared to the zebrafish ST8Sia II [49] favoring the idea of an acceptor substrate preference for the two fish enzymes and an evolution of their enzymatic properties. Similarly, the rainbow trout polysialyltransferases ST8Sia II and ST8Sia IV were shown to be involved the synthesis of PSA on both human N-CAM *N*-glycans and on the lake trout PSPG *O*-glycans [50,64]. In addition, these studies reported that even though the fish recombinant enzymes showed activity towards mammalian acceptor substrates used in enzymatic assays like bovine fetuin or human N-CAM, they also demonstrated lower enzymatic activity compared to their mammalian counterparts [64,65]. In any case, the nature of acceptor substrate of the zebrafish ST8Sia VI-like enzyme still awaits identification.

### 2.4. Molecular Phylogenetic and Phylogenomics Analyses Underscore Loss of the Teleost Fish st8sia6 Locus

It was then desirable to shed light into the evolutionary relationships of this zebrafish ST8Sia sequence with other mono-α2,8-sialyltransferases. Towards this aim, we conducted a combination of molecular phylogenetic (tree-based) and phylogenomic approaches. A total of 147 predicted α2,8-sialyltransferases sequences comprised of 129 vertebrate mono-α2,8-sialyltransferases (i.e., 17 ST8Sia I, 15 ST8Sia V, 63 ST8Sia VI and 34 ST8Sia VII) and 18 oligo-α2,8-sialyltransferases (i.e., ST8Sia III) identified in silico were used in multiple sequence alignments and the construction of phylogenetic trees. Figure 5 shows a phylogenetic tree obtained with the Neighbor-Joining (NJ) method in MEGA7.0 [66], rooted by the oligo-α2,8-sialyltransferases ST8Sia III. It indicates the presence of six groups of mono-α2,8-sialyltransferases. Intriguingly, the ST8Sia VI-related sequences split into two distinct sub-groups, one comprising only Teleost fish ST8Sia VI-like sequences the other comprising Chondrichthyes (sharks), basal ray-finned fishes like the gar *Lepisosteus oculatus* and Sarcopterygii (lobe-finned fish and tetrapods) ST8Sia VI sequences. Molecular phylogenetic analysis conducted by Maximum Likelihood or Minimum Evolution method also evidenced two disconnected groups of ST8Sia VI-related sequences (Appendix A). Furthermore, contrary to the ST8Sia VII sequences, the fish ST8Sia VI-like sequences form a tight group with short branches suggesting that these sequences could have evolved a new function distinct from the Chondrichthyes and Sarcopterygii ST8Sia VI sequences.

Sequence similarity networks [68] were also used as models to visualize evolutionary relationships between the various mono-α2,8-sialyltransferases and our data confirmed the phylogenic analyses (Figure 6). In the network, the most related sequences are grouped together in clusters. The greatest degree of similarity was found between sequences of the ST8Sia V, ST8Sia VI and the Teleost fish ST8Sia VI-like subfamilies, even at stringent cut-off values (1e-102). Interestingly, the tetrapod and shark ST8Sia VI sequences appear to be closer to the vertebrate ST8Sia V than the fish ST8Sia VI-like sequences. These last analyses casted doubt on the original nomenclature assigned to these fish ST8Sia VI-like sequences [31,35] and further suggested the molecular diversification of fish ST8Sia VI-related sequences among vertebrate mono-α2,8-sialyltransferases, and the possible occurrence of a novel *st8sia* gene subfamily in teleosts, therefore tentatively renamed ST8Sia VIII in Figure 6.

To assess this possibility, we also analyzed the genomic context for each gene locus and studied synteny and paralogy around the two *st8sia* genes (*st8sia6* and *st8sia8*) and adjacent gene loci in various vertebrate genomes, in the context of the two rounds of whole genome duplications (WGD-R1 and R2) that occurred at the base of vertebrates [69]. We observed conserved synteny around the *st8sia6* gene locus in the tetrapod genomes (human (*Homo sapiens*), mouse (*Mus musculus*) and chicken (*Gallus gallus*)) and in the fish genomes, although the *st8sia6* gene locus was lost in the spotted gar (*Lepisosteus oculatus*), zebrafish (*Danio rerio*) or medaka (*Oryzias latipes*) genome (Figure 7A). Of particular interest, the zebrafish *st8sia8* gene locus was identified in another conserved and distinct syntenic region found on *L. oculatus* LG15 and zebrafish chromosome 3. Neighboring genes *CACNB2B*, *ARL8*, *PLXDC2*, *NEBL/LASP* indicated in green in Figure 7A surrounding both *st8sia6* and *st8sia8* gene loci in the vertebrate genomes are paralogous, further indicating that the two *st8sia* genes share a common ancestor. Both loci are found on two different spotted gar linkage groups (LG9 and LG15) leading us to the conclusion that their common ancestor duplicated after the WGD-R2 (~500 MYA) and before the teleost genome duplication (TGD, ~360 MYA) event [70].

As another line of evidence, we further explored the relationships among these *st8sia*-related gene families taking advantage of the vertebrate [71] and chordate [72] ancestral genome reconstruction concept, previously described for the vertebrate *ST3GAL* genes [38]. As schematized in Figure 7B, 2R-duplicated genes are found on one of the 10 vertebrate ancestral chromosomes (VAC) in the pre-2R genome and designated a–j in the N-model [71]. Similarly, these genes are on one of the nine chordate linkage groups (CLG) in the pre-2R genome and named 1–9 in the P-model [72]. After the two WGD events, they are found on four linkage groups with shared synteny (*e.g*., Gnathostome ancestor (GNA) proto-chromosomes A0, A1, A2 and A3 in the N-model and chordate proto-chromosomes 1a, 1b, 1c and 1d in the P-model). Conserved synteny was established for *st8sia6* and *st8sia6*-like and surrounding gene loci and the blocks associated to these genes corresponded to GNA protochromosomes E1 and E2. Since the *st8sia7* gene locus was absent in the human, chicken and medaka genomes, we retrieved the corresponding block of synteny using a few neighboring genes like *eno3*, *ctc1* and *atp1b2a* widely distributed from fish to mammals. This genome reconstruction approach confirmed the scenario of a common origin of the *st8sia6* and fish *st8sia6*-like genes from a common ancestor after WGD-R2 illustrated in Figure 7C. Therefore, this new ray-finned fish mono-α2,8-sialyltransferase subfamily was definitively renamed ST8Sia VIII (*st8sia8* gene) according to the newly proposed sialyltransferase nomenclature [32]. In summary, the evolutionary relationships between the ST8Sia subfamilies were better resolved using the synteny/paralogy and paleogenomics approach than by the phylogeny methods only, which gave poorly resolved topologies.

## 3. Materials and Methods

### 3.1. Materials

The CMP-[^14^C]-Neu5Ac (10.7 GBq.mmol^−1^), ECL advance kit and First Strand cDNA Synthesis kit were from Amersham Pharmacia Biotech (Little Chalfont, UK). Enzymes Taq pol were from QBiogen. The Nucleospin RNA II kit was from Macherey-Nagel (Düren, Germany). Oligonucleotides were synthesized and purified by Eurogentec (Seraing, Belgium) and dNTP were from Promega Biosciences (Son Luis Obispo, CA, USA). DyNazyme Ext DNA polymerase was from Ozyme (Saint-Quentin-en-Yvelines, France). Dulbecco’s modified Eagle’s medium (DMEM) containing 4.5 g.l^−1^ glucose without glutamine was from BioWhittaker Europe. TC100 medium, minimal essential medium (MEM), L-glutamine, antibiotics, Geneticin G418, fetal calf serum used in cell culture, lipofectAMINE PLUS reagent and TOPO TA-cloning kit were from Invitrogen life technologies (Cergy-Pontoise, France); DMEM and Ultra MEM were from Lonza (Basel, Switzerland). Fetal bovine serum was from D. Deutscher (Issy-les-Moulineaux, France). N-Glycosidase F and anti-digoxigenin fluorescein Fab fragments were from Roche (Meylan, France). Hispeed Plasmid Midi kit was from Qiagen (Courtaboeuf, France). N-acetyl neuraminic acid (Neu5Ac), α1-acid glycoprotein, fetuin, 3xFLAG-CMV-9, 1,2-diamino-4,5methylenedioxybenzene dihydrochloride (DMB), arylgycosides and Triton CF-54 were purchased from Sigma-Aldrich (St. Louis, MO, USA). Glyco^®^ Sialidase S, Glyco^®^ Sialidase C and Glyco^®^ Sialidase A™ were from Glyko INC. (Novato, CA, USA). Polymerase chain reaction 8-well strip tubes, optical caps and 2X Brilliant^®^ SYBR^®^ Green Q-PCR master mix were from Stratagene (La Jolla, CA, USA). We synthesized 2-[4-({bis[(1-tert-butyl-1H-1,2,3-triazol-4-yl)methyl]amino}methyl)-1H-1,2,3-triazol-1-yl]acetic acid (BTTAA) in our laboratory as previously described [75]. NMR experiments were carried out on a Brüker Avance II 400 MHz NMR spectrometer equipped with a 5 mm BBO (^31^P, ^13^C) or a 5 mm TBI probe (^1^H). Deuterated solvents were purchased from Eurisotop. Chemical shifts were referenced to tetramethylsilane (^1^H, ^13^C) and phosphoric acid (^31^P) and are reported as part per million (ppm). Scalar J coupling constants are reported in hertz (Hz).

### 3.2. In Silico Identification and Analysis of ST8Sia Sequences in Databases

A BLAST search approach was used to retrieve the vertebrate *st8sia6* nucleotide sequences with significant homology to the previously described mammalian sequences (human ST8Sia VI AJ621583 [41] and mouse ST8Sia VI AB059554 [42], from the genomic and TSA divisions of the GenBank^®^/EBI databases at the National Center for Biotechnology Information (NCBI) [31,35]. The amino acid sequence analysis was performed using the software of Expert Protein Analysis System (ExPASy; Swiss Institute of Bioinformatics, Switzerland; website (https://www.expasy.org/, accessed on: 30 January 2019). Hydropathy analyses and determination of potential N-glycosylation sites were performed using the servers TM-Pred Prediction of Transmembrane Regions and orientation (https://embnet.vital-it.ch/software/TMPRED_form.html, accessed on: 30 January 2019) and the NetNGlyc 1.0 Internet program (http://www.cbs.dtu.dk/services/NetNGlyc/, accessed on: 30 January 2019) of ExPaSy. Sequence alignments were performed using the clustalW algorithms at the PRABI website (https://npsa-prabi.ibcp.fr/cgi-bin/npsa_automat.pl?page=/NPSA/npsa_clustalw.html, accessed on: 30 January 2019). Phylogeny was determined aligning the known vertebrate ST8Sia sequences with MUSCLE in MEGA7.0 [66].

### 3.3. Phylogenetic Analysis and Sequence Similarity Network

The multiple sequence alignment of 147 selected vertebrate ST8Sia sequences (i.e., 17 ST8Sia I, 15 ST8Sia V, 63 ST8Sia VI and 34 ST8Sia VII and 18 oligo-α2,8-sialyltransferases (ST8Sia III) used as outgroup) was conducted using MUSCLE and Clustal Omega algorithms included in MEGA7.0 software and refined by hand (see Appendix A). Phylogenetic trees were produced by the Neighbor-Joining (NJ), Maximum Likelihood and Minimum Evolution method in MEGA 7.0 [66,67].

Sequence similarity network was constructed using formatdb from the standalone BLAST software and a custom BLAST database was created using the same set of sequences used for the phylogenetic analysis. A graphical overview of interrelationships among and between sets of proteins was provided at different E value thresholds as previously described [38]. The network was visualized using Cytoscape [76] where the node represents ST8Sia sequences and edges are defined between any pair of nodes with an E value less than the threshold (1e-95, 1e-100, 1e-102). Nodes were colored according to the subfamily to which the sequence belongs.

### 3.4. Synteny Analysis, Paralogon Detection and Ancestral Genome Reconstruction

Synteny between the *st8sia* gene loci and vicinal genes in vertebrate genomes was assessed by manual chromosome walking and reciprocal BLAST searches. Detection of paralogous blocks was achieved and visualized using the Genomicus site (version 92.01) http://www.genomicus.biologie.ens.fr/genomicus-92.01/cgi-bin/search.pl, last accessed August 2018 [73]. When the *st8sia* gene of interest was absent in a genome, we used genes physically close as a seed to identify syntenic segments. Ancestral genome reconstruction was used in conjunction with phylogenetic and synteny analysis to rapidly assess the dynamic of *st8sia* genes evolutionary relationships, taking into account the reconstruction of proto-chromosomes of ancestral vertebrates [71,74] and of chordate [72] as previously reported [38,77].

### 3.5. RNA Extraction, cDNA Synthesis and RT-PCR

Total RNA was extracted from various zebrafish adult tissues and embryos (0, 6, 14, 24 and 36 hpf) using the Qiagen RNeasy kit, and cellular RNA was quantified by spectrophotometry using the NanoDrop^®^ ND-1000 spectrophotometer (NanoDrop Technologies, Wilmington, DE, USA). The integrity and purity of the extracted RNA was also analyzed by means of gel electrophoresis on a bioanalyzer (Experion, Bio-Rad Laboratories, Inc, Marnes-la-Coquette, France). For subsequent PCR amplifications, first-strand cDNA was synthesized from total RNA using the First Strand cDNA Synthesis kit according to the manufacturer’s protocol in a final volume of 33 µL. A specific zebrafish *st8sia6*-like fragment of 333 bp was obtained after RT-PCR of RNAs isolated from various adult and embryonic tissues using 35 nM of sense (5’-TGTCTATGATGGCGAAAG-3′) and antisense (5′-TGACCGTATGAATGAAGG- 3′) primers, 100 µM of dNTP and 0.5 unit of DNA Taq polymerase using the following conditions: 96 °C for 2 min, 38 cycles of 45 sec at 95 °C, 50 sec at 50 °C and 1 min at 72 °C, and 10 min at 72 °C. Expression of the *Dreβ-actin* gene was followed in the same RT-PCR conditions as a control of cDNA synthesis and purity. For that purpose, *Dreβ-actin* sense primer (5′_132_GTTGGTATGGGACAGAAAGA3′) and antisense primer (5′_509_GGCGTAACCCTCGTAGAT3′) were designed in two different exons of the zebrafish *β-actin* gene (Accession number AF025305) and synthesized by Eurogentec. The RT-PCR products were subjected to 2% agarose gel electrophoresis and amplification of cDNA resulted in a 378 bp fragment.

### 3.6. Isolation of A Zebrafish st8sia6 cDNA and Construction of an Expression Vector

To obtain a cDNA encoding the full-length protein, RT-PCR was performed using 1 µg of the phage kidney oligo d(T) primed cDNA library kindly provided by L. Zon (Boston Children Hospital, Boston, MA, USA). A first cDNA amplification was performed by PCR using a sense (5′_−246_ AGAGCGGCAGCATCTG 3′) and an antisense (5′_1492_ CATTTCCCACCAGCCTCGT 3′) zebrafish *st8sia6* specific oligonucleotide primers, at 95 °C for 2 min, followed by 38 cycles (96 °C for 45 sec, 53 °C for 1 min and 72 °C for 90 sec), and an extension step of 10 min at 72 °C. The RT-amplified fragments (1183 bp) were subcloned into PCR2.1 TOPO TA cloning vector. For subsequent plasmid constructions, restriction digestion and DNA sequencing (GATC Biotech, Germany) confirmed the insert junctions. Several expression vectors were prepared for subsequent transient transfection in animal COS-7or HEK293 cells. A cDNA encoding the full length form of *Dre* ST8Sia VI-like enzyme was obtained by PCR amplification using the previous construct as DNA template, the sense primer containing the restriction site EcoRI (5′-GACCGTGTGAATTCGTGGATGCGGGTTATGAG-3′) and the antisense primer containing the restriction site KpnI (5′-TCCCACCAGGTACCTGTTTCATCTATGAGCGG-3′). Reactions were run for 2 min at 96 °C followed by 35 cycles (94°C for 45 sec and 1 min at 72 °C) and an extension step of 10 min at 72°C. The resulting PCR fragment was subcloned into the pCR2.1 vector of TOPO TA-cloning kit, was cut out by digestion with EcoRI and KpnI, and inserted into the EcoRI and KpnI sites of the 3xFLAG-CMV-10 expression vector. The resulting plasmid encoded a fusion protein with the FLAG sequence and the full length form of the ST8Sia VI-like sequence. This full length cDNA was also inserted into a pRC-CMV vector encoding the full length form of the ST8Sia VI-like sequence with no tag. A cDNA encoding a truncated form of *Dre* ST8Sia VI-like lacking the first 32 amino acids of the open reading frame was also obtained by PCR amplification using the same DNA template, the sense primer containing the restriction site EcoRI (5′-CATCTCCAAGAATTCTGTAATCCCTCATCCTGC-3′) and the antisense primer containing the restriction site KpnI (5′-TCCCACCAGGTAC**C**TGTTTCATCTATGAGCGG-3′). Reactions were run for 2 min at 95 °C followed by 38 cycles (96 °C for 45 s, 47 °C for 1 min and 90 s at 72 °C) and an extension step of 10 min at 72 °C. The resulting 1015 bp fragment was subcloned into the pCR2.1 vector of TOPO TA-cloning kit, was cut out by digestion with EcoRI and KpnI, and inserted into the EcoRI and KpnI sites of the 3xFLAG-CMV-9 expression vector. The resulting plasmid encoded a fusion protein with a signal peptide sequence, the FLAG sequence, and a truncated form (Δ33ST8Sia VI-like) lacking its cytoplasmic and transmembrane domain.

### 3.7. Animals, Cell Culture and Transient Expression of a Soluble form of Dre ST8Sia VIII

The zebrafish, *D. rerio*, were maintained in aquaria at 28 °C as described previously [78] and the day-night cycle was controlled with an automated timer (14 h light/10 h dark). Experiments were performed using random matings of AB animals and all experimental procedures adhered to the CNRS (Centre National de la Recherche Scientifique). Embryos were collected, raised and staged in embryos medium until they reached the desired developmental stage determined following previously defined criteria [79].

COS-7 (ATCC CRL-1651) or HEK293 (ATCC CRL-1573) cells were grown in DMEM medium supplemented with 10% FCS, L-glutamine 20 mM, Penicillin, Streptomycin at 37 °C under 5% CO_2_. Confluent cells (70%) were transiently transfected using 5 µg of purified 3xFLAG-CMV-9 ST8Sia VI-like or 3xFLAG-CMV-9 in 100-mm diameter dishes using Lipofectamine PLUS reagent, following the manufacturer’s instructions. The transfected cells and culture media were harvested 48 h after transfection, and the recombinant ST8Sia VI-like enzyme expressed both in the cell culture medium and within the transfected cells were used as crude enzyme source for enzymatic activity assays.

### 3.8. Synthesis of Activated CMP-Sialic Acid



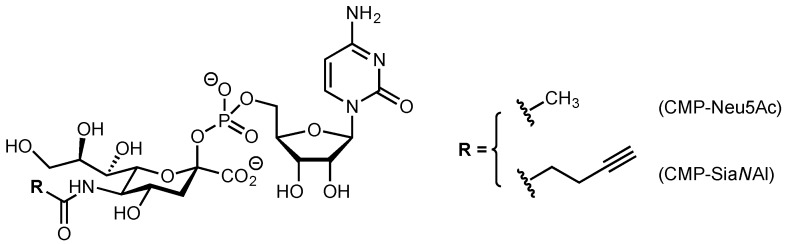



*N*-(4-pentynoyl)neuraminic acid (Sia*N*Al) was synthesized from commercial D-mannosamine hydrochloride as previously described [80]. Cytidine-5′-monophospho-*N*-(4-pentynoyl)neuraminic acid (CMP-Sia*N*Al) and Cytidine-5′-monophospho-*N*-acetylneuraminic acid (CMP-Neu5Ac) were freshly prepared prior to use according to our reported chemo-enzymatic procedure. Synthesis of CMP-Sia*N*Al was carried out as previously described [43]. In brief, the relevant sialic acid (1 eq.) and cytidine 5′-triphosphate disodium salt (10.6 mg, 0.02 mmol, 1 eq.) were dissolved into 650 µL of Tris-HCl buffer (100 mM, pH 8.5) containing 20 mM MgCl_2_ in a 5 mm NMR tube. When necessary, the pH of the solution was readjusted to 8.5 using aqueous ammonia prior to adding the enzymes. We then added to the mixture 0.3 U of CMP-Sialic acid synthetase from *Neisseria meningitidis* group B (Sigma Aldrich, EC 2.7.7.43), and 0.5 U of inorganic pyrophospatase from *Saccharomyces cerevisiae*, (Sigma Aldrich, EC 3.6.1.1). The sample was inserted in a Brüker Avance II 400 MHz NMR spectrometer equipped with a 5 mm BBO probe set for the ^31^P nucleus resonance frequency and temperature-regulated at 37 °C, and the reaction was monitored by 1D ^31^P NMR. Upon completion, the solution was cooled down to 4 °C and directly diluted to the right concentration for subsequent sialylation assays with no further purification. CMP-Sia*N*Al was characterized by ^1^H, ^13^C and ^31^P NMR. NMR ^1^H (400 MHz, D_2_O): δ = 7.51 (d, J = 7.6, 1H), 5.72 (d, J = 7.6, 1H), 5.58 (d, J = 3.9, 1H), 4.00–3.87 (m, 4H), 3.85 (d, J = 5.5, 1H), 3.76 (d, J = 10.4, 1H), 3.74–3.66 (m, 1H), 3.61 (d, J = 10.3, 1H), 3.56 (dd, J = 10.2, 3.0, 1H), 3.51 (d, J = 12.2, 1H), 3.26 (dd, J = 14.2, 7.1, 1H), 3.20 (dd, J = 10.8, 6.0, 1H), 2.22–2.03 (m, 5H), 1.99 (s, 1H), 1.26 (ddd, J = 13.0, 11.7, 5.8, 1H). ^13^C (101 MHz, D2O): δ = 175.24, 174.38, 165.92, 157.54, 141.42, 99.90, 96.53, 88.98, 83.22, 82.31, 73.94, 71.65, 70.31, 69.18, 69.01, 68.78, 66.57, 64.75, 62.90, 51.63, 40.98, 34.65, 14.56. ^31^P (162 MHz, D_2_O): δ = −4.63.

### 3.9. Enzymatic Characterization of Sialyltransferase

Sialyltransferase assays were performed in 100 mM cacodylate buffer, pH 6.2 containing 10 mM MnCl_2_, 0.2% Triton CF-54, 40 µM CMP-[^14^C]Neu5Ac (1.94 KBq) and one of the acceptor substrates (2 mg.mL^−1^ for glycoproteins) and 23 µL of the enzyme source in a total volume of 50 µL as previously described [41]. The recombinant CHO-derived human Δ27ST8Sia VI (R & D systems Europe Ldt, France) was used as a control as previously described [41]. Unless stated otherwise, the reactions were performed at 27 °C for 4 h. For glycoproteins, the reactions were stopped by adding 2.5X SDS/sample buffer and the reaction products were separated on SDS-PAGE. After transfer onto a nitrocellulose membrane (Biotrace, Pall corporation, Ann Arbor, MI, USA), the radioactive products were detected and quantified by radio-imaging using a Personal Molecular Imager FX (Bio-Rad, France). Sialyltransferase assays were also performed using the MPSA described recently [45]. Briefly, 400 ng of glycoprotein acceptor in 100 µL of sodium bicarbonate buffer (20 mM pH 9,6) or glycolipids mixture were coated in 96-well plate wells (F8 MaxiSorp Loose Nunc-Immuno Module ThermoScientific) overnight at 4 °C. After three washes with 150 µL of PBST-0.05% (Phosphate Buffer Saline-Tween) saturation was carried out for 1 h at room temperature using 100 µL of oxidized BSA at 0.05% dissolved in sodium bicarbonate buffer. After incubating for 3 h with the enzymatic source, 1 mM CMP-Sia*N*Al and cacodylate buffer in a total volume of 50 µL, the wells were washed with PBST-0.05% and 100 µL of a CuAAC labeling solution containing 300 µM of CuSO4, 600 µM of BTTAA, 2.5 mM of sodium ascorbate and 250 µM of azido-biotin in PBS was then added. After 1 h incubation at 37 °C, washes were made with PBST-0.05% and wells were incubated with 100 µL of anti-biotin antibody HRP-conjugated diluted to 1/25,000 in PBST-0.05% for 1 h at 37 °C. After washes, 100 µL of TMB (3,3′,5,5′-tetramethylbenzidine) was added and incubated 20 min at room temperature in the dark. Finally, absorbance was quantified at 620 nm using spectrophotometer (SpectroStar Nano; BMG Labtech).

### 3.10. Real Time PCR

Real-time PCR and subsequent data analysis were performed using the Mx4000 Multiplex quantitative PCR System (Stratagene) equipped with version 3.0 software. Each 25 µL PCR reaction contained 12.5 µL of the 2X Brilliant SYBR^®^ Green Q-PCR mix, 150 nM of each primer (sense (5’-TGTCTATGATGGCGAAAG-3′) and antisense (5′-TGACCGTATGAATGAAGG- 3′) primers) and 2 µL of cDNA diluted to 1/20 (100 ng). DNA amplifications were performed in triplicates from two biological samples with the following thermal cycling profile: initial denaturation at 95 °C for 10 min, 45 cycles of amplification (denaturation 95 °C for 30 sec, annealing at 50 °C for 1 min and extension at 72 °C for 30 s) and a final extension at 72 °C for 5 min and this was followed by a melting step consisting of heating from 50 to 95 °C at an increment of 1 °C per 30 sec to check the specificity of the amplified product. The fluorescence monitoring occurred at the end of each cycle. The reactions were quantified by selecting the amplification cycle when the PCR product of interest was detected (threshold cycle, Ct). Calibration curves were generated by 10-fold serial dilution of HindIII linearized TOPO plasmids containing the amplified regions of the targeted gene.

### 3.11. In Situ Hybridization

Antisense DIG labeled RNA probe synthesis and whole-mount *in situ* hybridization were performed according to [81]. The fully detailed protocol is accessible at http://zfin.org/zf_info/zfbook/chapt9/9.82.html. Briefly, for synthesis of antisense ST8Sia VI riboprobe, the zebrafish ST8Sia VI-like-containing PCR II-TOPO plasmid was linearized by digestion with *Xba*I. Digoxigenin-labeled antisense riboprobe was synthesized by in vitro transcription using T7 polymerase (Promega). For double labeling with NBT/BCIP and Fast Red (Roche), the hybridized embryos were first incubated with pre-absorbed anti-digoxigenin antibody (2–3 h; Roche), followed by staining with NBT/BCIP. The staining reaction was stopped by two 5-min washes in PBST, 10 min with 0.1 M Glycine-hydrochloric acid (pH = 2.2) with 0.1% Tween 20 and four 5-min washes in PBST. The washed embryos were incubated with the pre-absorbed anti-fluorescein antibody (Roche), washed and stained with Fast Red-staining solution (two tablets dissolved in 4 mL of 0.1 M Tris-HCL, pH 8.2, with 0.1% Tween 20). Stained embryos were rinsed in PBST and post-staining fixed in 4% PFA before subject to imaging. Images of the stained embryos were acquired with a Leica MZ FLIII Stereomicroscope and a magnifier cooled CCD camera. Specificity was assessed using antisense and other irrelevant probes (data not shown).

## 4. Conclusion

In this study, we have identified and characterized a new zebrafish α2,8-sialyltransferase sequence in *Danio rerio*. Although this enzyme needs to be better kinetically characterized, we showed that it has almost no transfer activity on sialylated fetuin. Furthermore, our molecular phylogeny data demonstrate that this sequence is not the fish orthologue of the mammalian ST8Sia VI described previously [41,42]. This sequence belongs to a new teleost mono-α2,8-sialyltransferase subfamily resulting from the second whole genome duplication event. This newly described subfamily renamed ST8Sia VIII was maintained in ray-finned fish and disappeared in Chondrichthyes and Sarcopterygii, whereas the ST8Sia VI subfamily disappeared in ray-finned fish and was maintained in Chondrichthyes and Sarcopterygii.

## Figures and Tables

**Figure 1 ijms-20-00622-f001:**
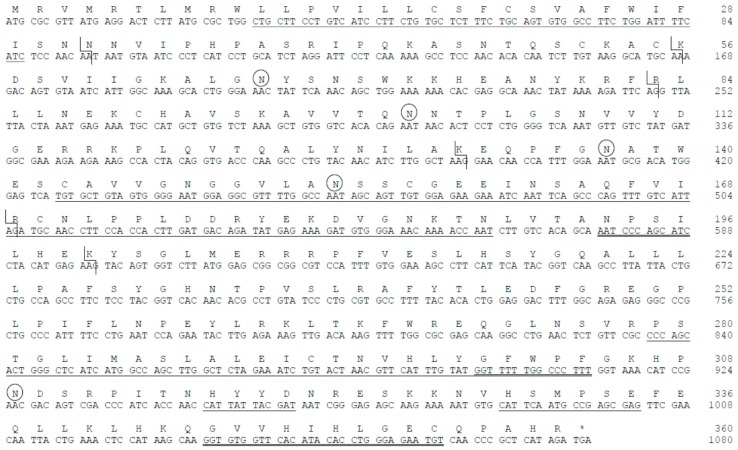
Nucleotide and predicted amino acid sequences of the zebrafish ST8Sia VI-like (*Dre*ST8Sia VI). Numbering of the cDNA begins with the initiation codon. The amino acid sequence is shown in single-letter code. The putative 19 amino acid N-terminal transmembrane domain is underlined with a grey line and the putative *N*-glycosylation sites (N-X-S/T) are circled. The sialylmotifs L, S, III and VS are underlined with a single black line and the ST8Sia family motifs with a double black line. The broken line in the nucleotide sequence indicates the exon/intron junctions.

**Figure 2 ijms-20-00622-f002:**
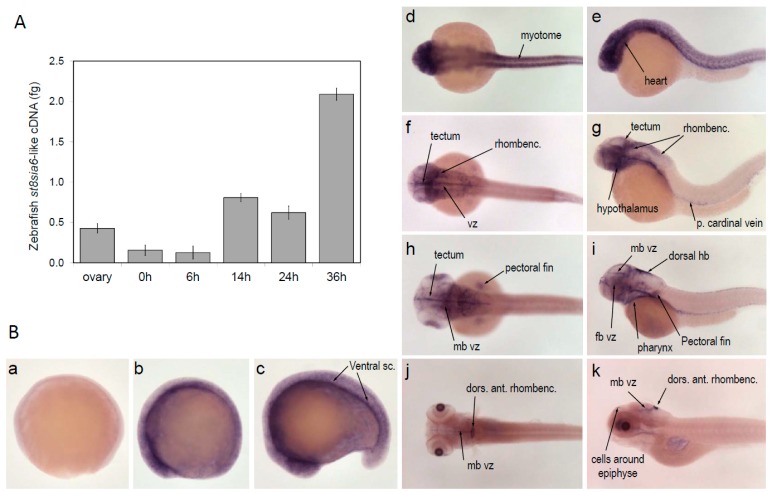
Zebrafish *st8sia6*-like gene spatio-temporal expression during embryonic and larval development. (**A**) Absolute quantification of the mRNA levels of the zebrafish *st8sia6*-like gene was achieved by Q-PCR in ovary, and at 0, 6, 14, 24 and 36 hpf. PCR for two biological samples was carried out in triplicate as described in the materials and methods section, and data are expressed as means +/-SD. (**B**) The zebrafish *st8sia6*-like gene distribution was studied by whole mount in situ hybridization with a digoxigenin-labeled Dre *st8sia6*-like antisense riboprobe. Developmental stage is given as hpf and anterior is to the left: (**a**) gastrula stage (5 hpf); (**b**) early somitogenesis (10 hpf); (**c**) mid-somitogenesis (15 hpf), (**d**,**e**) lateral and dorsal views at pharyngula stage (24 hpf), (**f**,**g**), lateral and dorsal views at 36 hpf, (**h**,**i)** hatching time (48 hpf), (**j**,**k**), larva stage (120 hpf).

**Figure 3 ijms-20-00622-f003:**
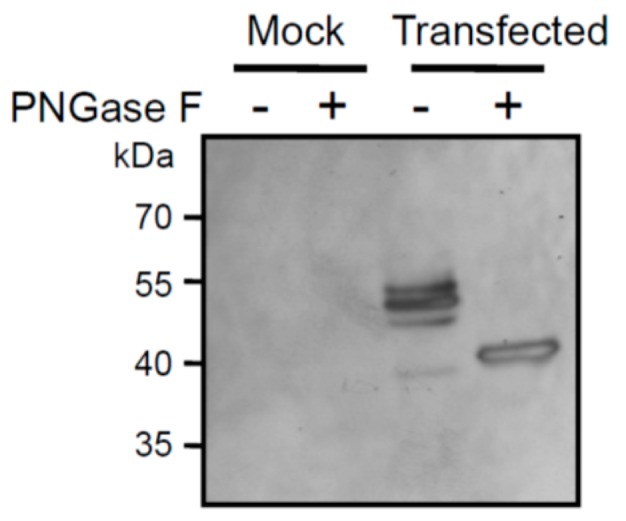
Immunoblotting of the zebrafish ST8Sia VI-like recombinant protein produced in transfected COS-7 cells media. COS-7 cell culture media from pFLAG-CMV-9/Dre ST8Sia VI-like or mock transfected cells (48 h post-transfection) were treated (+) or not (−) with PNGase F and were subjected to SDS-PAGE under reducing conditions and Western blotting using the BioM2 anti-FLAG mAb. The position of the high range pre-stained SDS-PAGE standards are indicated in kDa on the left side. Lane 1 and 2: mock transfected cells; Lane 3 and 4: pFLAG-CMV-9/DreST8Sia VI-like transfected cells.

**Figure 4 ijms-20-00622-f004:**
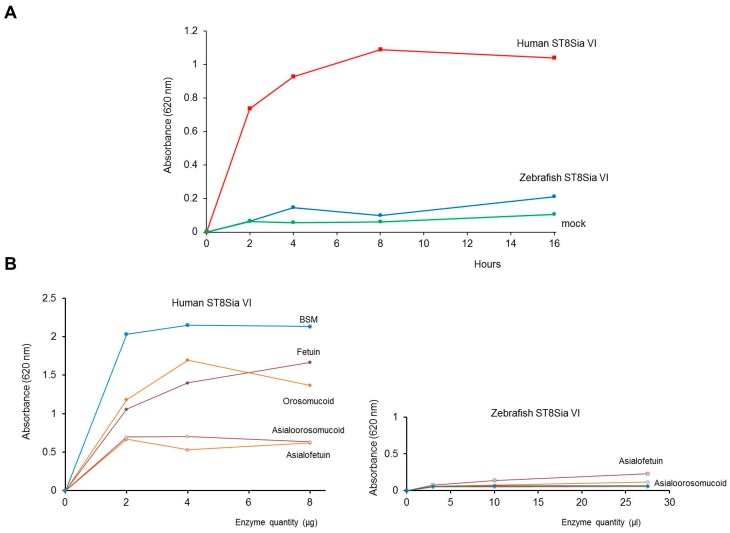
Enzymatic characterization of the ST8Sia VI enzymes using the MicroPlate Sialyltransferase Assay (MPSA). (**A**) Time course of human and zebrafish ST8Sia activities onto bovine fetuin. Sialylation reactions were conducted at 25 °C for various incubation times (2–16 h) and 100 µM CMP-Sia*N*Al with either pFLAG-CMV-9/Dre ST8Sia VI-like or mock HEK293-transfected cells (27.5 µL) or 8 µg of recombinant human ST8Sia VI, *n* = 2. (**B**) Human and zebrafish ST8Sia activities on various mammalian glycoproteins using the MPSA. Sialylation reactions were conducted for 4 h at 27 °C using potential acceptor substrates (fetuin, asialofetuin, orosomucoid, asialoorosomucoid and bovine submaxillary mucin (BSM) and 100 µM CMP-Sia*N*Al with variable amounts (3, 10, 27.5 µL) of cell culture media from pFLAG-CMV-9/Dre ST8Sia VI-like or mock HEK293-transfected cells or recombinant human ST8Sia VI (1–8 µg). Data shown on the left side are those corresponding to the human enzyme and those on the right side are those of the zebrafish enzyme, *n* = 2.

**Figure 5 ijms-20-00622-f005:**
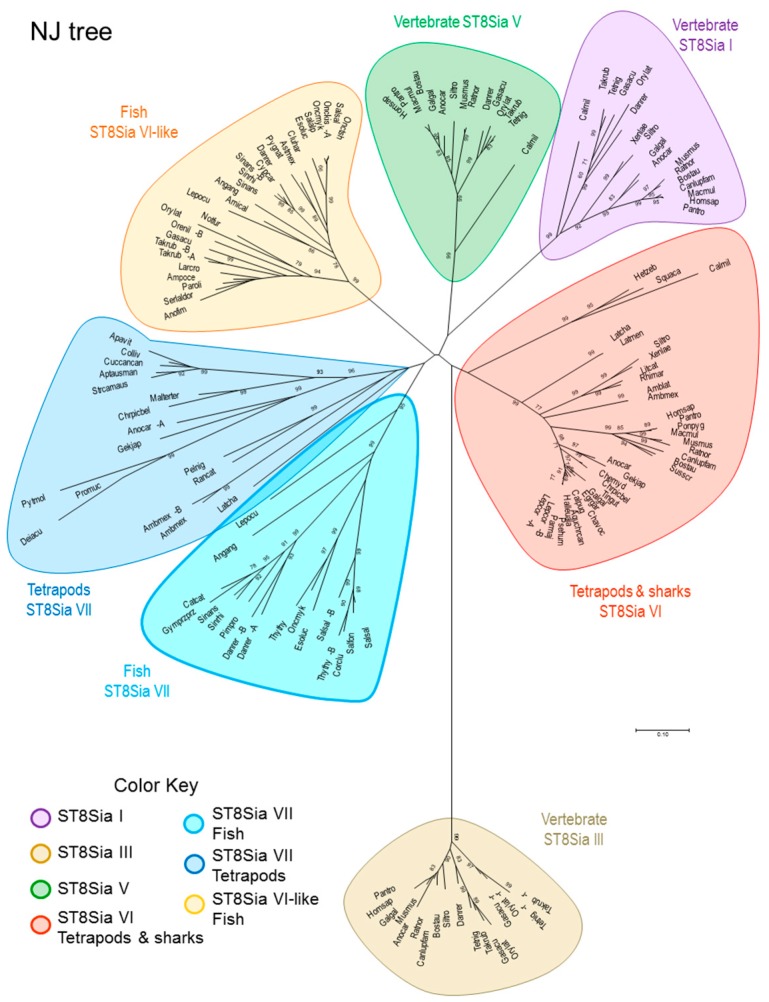
Unrooted Neighbor-Joining (NJ) phylogenetic tree showing the evolutionary relationships between the zebrafish ST8Sia VI-like sequence and the other vertebrate mono- and oligo-α2,8-sialyltransferases of the ST8Sia family. Amino acid sequences of 147 selected vertebrate ST8Sia sequences (i.e., 17 ST8Sia I, 15 ST8Sia V, 63 ST8Sia VI-related and 34 ST8Sia VII and 18 oligo-α2,8-sialyltransferases (ST8Sia III) used as outgroup). Multiple sequence alignment was conducted using MUSCLE in MEGA 7.0 and refined by hands. Phylogenetic trees were produced by the NJ method in MEGA 7.0 [66,67].

**Figure 6 ijms-20-00622-f006:**
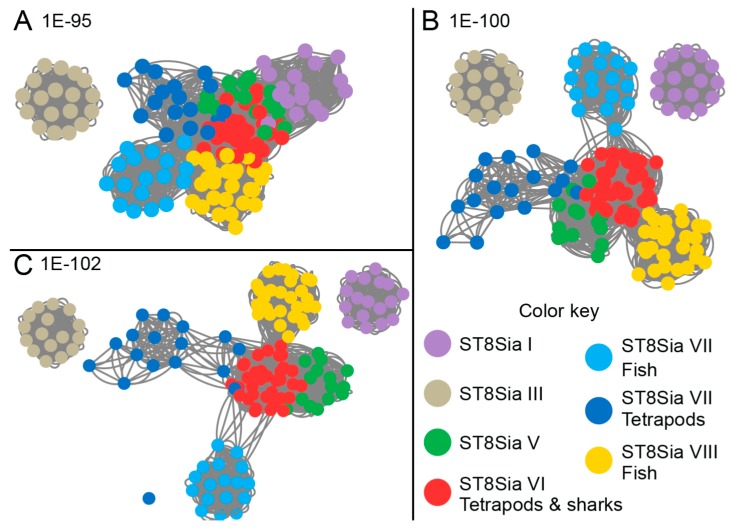
ST8Sia sequence similarity network. The sequence similarity network including 147 vertebrate ST8Sia sequences was constructed as described in material and methods section. The sequences are represented as nodes colored according to the subfamily to which they belong. The lines between two nodes are drawn if the BLAST E-value is below the threshold indicated in the figure. Three E-value thresholds are used to visualize the interconnectivity evolution, ordered.

**Figure 7 ijms-20-00622-f007:**
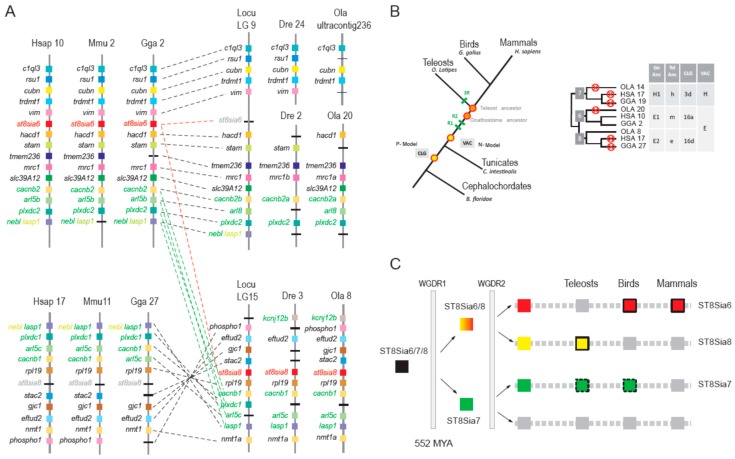
Evolutionary history of the *st8sia6* and *st8sia8* gene loci. (**A**) Syntenic relationships of the *st8sia6* and *st8sia8* gene loci in vertebrates. Chromosomal locations of the *st8sia6* and *st8sia8* and neighboring gene loci were determined in the human (*Homo sapiens*, Hsa), the mouse (*Mus musculus*, Mmu), the chicken (*Gallus gallus*, Gga), the spotted gar (*Lepisosteus oculatus*, Locu), the zebrafish (*Danio rerio*, Dre) and the medaka (*Oryzias latipes*, Ola) genomes. Putative orthologues were determined with information from the NCBI and ENSEMBL servers and visualized using the Genomicus 93.01 web site [73]. Paralogous genes in the vertebrate genomes are indicated in green, the *st8sia6* and the fish *st8sia6*-like (i.e., *st8sia8*) gene loci are indicated in red or in grey when lost in the considered genome. (**B**) Reconstruction of ancestral genome to assess ST8Sia VIII subfamily origin. A schematic phylogenetic tree is shown on the left side to illustrate evolution of the chordate genome using the N-model [71,74] for the reconstruction of the prevertebrate ancestor (VAC, N-model) and the P-model [72] for the reconstruction of the prechordate ancestor (CLG, P-model). On the right side is a schematic illustration of the data obtained using the known genomic location of *st8sia* genes in *O. latipes* (OLA), in *G. gallus* (GGA) and in *H. sapiens* (HSA). Crosses indicate gene losses, Tel Anc indicates pre-3R teleost ancestor and Gn Anc indicates post-2R Gnathostome ancestor. (**C**) Schematic diagram illustrating the *st8sia6/7/8* gene locus evolution after the two whole genome duplication (WGD) rounds. The ancestral gene *st8sia 6/7/8* indicated by a black box gave two duplicates, the ancestral *st8sia6/8* and *st8sia7* genes after the WGD-R1, 552 MYA [35]. Four duplicated genes have arisen after the WGD-R2 event and *st8sia6, st8sia8, st8sia7* were maintained in vertebrate genomes. The *st8sia6* gene was lost in the Actinopterygii (ray-finned fish) genome and maintained in Sarcopterygii (lobe-finned fish and tetrapods), whereas the *st8sia8* was lost in Sarcopterygii and maintained in Actinopterygii. A disrupted line frames the *st8sia7* green box to indicate that this gene is not found in all teleost or bird species. Grey boxes indicate that the gene was lost in the branch.

**Table 1 ijms-20-00622-t001:** Sequence similarity analysis of the zebrafish ST8Sia VI-like. The zebrafish ST8Sia VI-like sequence was compared to the 20 known human sialyltransferase sequences.

*Homo sapiens*	Accession Number	*Danio rerio* ST8Sia VI-like (AJ715551)
ST8Sia I	D26360	33.3%
ST8Sia II	U33551	29.7%
ST8Sia III	AF004668	29.9%
ST8Sia IV	L41680	27.7%
ST8Sia V	U91641	35.8%
ST8Sia VI	AJ621583	35.8%
ST3Gal I	L29555	21.4%
ST3Gal II	X96667	23.7%
ST3Gal III	L23768	20.3%
ST3Gal IV	L23767	23.6%
ST3Gal V	AB018356	24.8%
ST3Gal VI	AF119391	22.1%
ST6Gal I	X17247	19%
ST6Gal II	AB059555	17.2%
ST6GalNAc I	Y11339	16%
ST6GalNAc II	AJ251053	23.6%
ST6GalNAc III	AJ507291	16.9%
ST6GalNAc IV	AJ271734	20%
ST6GalNAc V	AJ507292	20.5%
ST6GalNAc VI	AJ507293	19.4%

**Table 2 ijms-20-00622-t002:** Sequence similarity analysis of the ST8Sia orthologues. The human ST8Sia sequences were compared to their zebrafish orthologues.

Sialyltransferase	*Homo sapiens*	*Danio rerio*	Identity (%)
ST8Sia I	D26360	AJ715535	58.7%
ST8Sia II	U33551	AY055462	60.9%
ST8Sia III	AF004668	AJ15541	77.9%
ST8Sia IV	L41680	AJ715545	67.8%
ST8Sia V	U91641	AJ715546	72.1%
ST8Sia VI	AJ621583	AJ715551	35.8%

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
