# Peer review of "Novel Zebrafish Mono-α2,8-sialyltransferase (ST8Sia VIII): An Evolutionary Perspective of α2,8-Sialylation"

_ijms, 2019, doi:10.3390/ijms20030622_

Round 1
Reviewer 1 Report
@page { margin: 0.79in } p { margin-bottom: 0.1in; direction: ltr; line-height: 115%; text-align: left; orphans: 2; widows: 2 }
I believe this manuscript will be well received in the special issue “Glycan-Receptor Interaction” of IJMS and I greatly appreciate the evolutionary approach to investigating enzyme function. In this paper, the authors took advantage of well-established genomes (ie zebrafish, human and mouse) and compared gene sequences to more recent genome efforts (such as spotted gar) in order to define the gene locus for ST8Sia6 and suggest renaming fish-ST8Sia6 to fish-ST8Sia8. The need to adjust this gene name in fishes was further solidified by showing functional differences exist between human ST8Sia6 and the renamed fish-ST8Sia8.
I have one concern over the validity of the functional assay that likely stems from my ignorance of ST8Sia enzymes. The assays used in the manuscript used human ST8Sia6 to measure the amount of SiaNAl transfer activity to variety of mammalian substrates as a positive control. They then showed that zebrafish ST8Sia6 (now renamed ST8sia8) had no SiaNAl transfer activity on the same mammalian substrates. Can the authors provide proof (perhaps just a reference), that fish enzymes would be expected to operate on mammalian substrate? The reciprocal would also reduce my concern – can the authors show that human ST8Sia6 has activity on fish substrates? It seems like the authors acknowledge this in line 275 “These data suggested a loss of function of this enzyme or a low affinity for the used mammalian acceptor substrates...” but is immediately ignored in the next sentence line 276: “Altogether, these data show that the MPSA approach could be used to determine […] activities onto the glycoprotein acceptors” - I don’t believe that statement.
Overall comments:
Figure legends: There seems to be an issue with all the figure legends where there are two titles. This is likely an auto-formatting issue from the upload to the journal submission; however some of the repeated titles are not 100% identical - just pointing this out so the editor or authors can pick the correct title. In addition, the figure legends contain too much detail regarding the methods. There is a "MATERIALS AND METHODS" section and this information is also appropriately found there. In addition, there are results and conclusions found in many of the figure legends – which is also more appropriately found in the “RESULTs and DISCUSSION” section and should be removed from the figure legends.
Figure 2. (A) y axis could use a more informative label – “Zebrafish st8sia6-like cDNA (fg)” perhaps? “Q-PCR” is usually written with a lower case “q”.
Figure 4. B is described before A is. Should be switched.
Figure 5. The resolution of my copy of the figure is not high enough for me to read the names – even when viewing the electronic copy. I’m not sure what the coloring scheme means – why are most species names (which I can’t read most of them) blue for the gene families, but ST8Sia V and I are green or purple? If the resolution can’t be improved, then perhaps each species name could be colored and marked in a key or maybe a couple arrows pointing to specific points?
Table legends: Remove methods from the table legends. This amount of captioning is unusual and is better served in the “MATERIALS AND METHODS” section. Table 1 and Table 2 have the exact same name.
Minor grammatical comments/suggestions:
Line 26: “MPSA sialylation assay” suffers from RAS syndrome (redundant acronym syndrome syndrome). I suggest spelling out the entire acronym in a the abstract, since this is the first time readers will encounter it.
Line 49: New paragraph beginning with “In teleost fishes…”
Line 57: “,most of these sialylated structures comprising” is a sentence fragment and needs reworded
Line 81: “substrates” to “substrate”
Line 101: this sentence is unclear. It almost reads like snakes and lizards are teleost fishes…
Line 129: add “nucleotide” in front of “sequence”
Line 129: change “seed” to “query”
Line 129: The BLAST source is very old and is for a protein specific search. PMID: 20003500 is probably more appropriate
Line 131: “exhaustive searches in transcriptomic databases” cites a single database. This does not seem
Line 140: “which locations are not conserved neither in…” is awkward.
Line 171: “quantitatively” is redundant.
Line 189: Sentence starting with “At larval stage” needs some rewording
Line 234: suggest changing “admitted” to something else such as “accepted”
Line 258: “We, therefore” - remove the comma
Line 455: Sources 32 and 35 don’t seem to fit here
Line 425 and 126 - lowercase vs capitalization difference in headers with "and"
Author Response
I believe this manuscript will be well received in the special issue “Glycan-Receptor Interaction” of IJMS and I greatly appreciate the evolutionary approach to investigating enzyme function. In this paper, the authors took advantage of well-established genomes (ie zebrafish, human and mouse) and compared gene sequences to more recent genome efforts (such as spotted gar) in order to define the gene locus for ST8Sia6 and suggest renaming fish-ST8Sia6 to fish-ST8Sia8. The need to adjust this gene name in fishes was further solidified by showing functional differences exist between human ST8Sia6 and the renamed fish-ST8Sia8.
I have of one concern over the validity the functional assay that likely stems from my ignorance of ST8Sia enzymes. The assays used in the manuscript used human ST8Sia6 to measure the amount of SiaNAl transfer activity to variety of mammalian substrates as a positive control. They then showed that zebrafish ST8Sia6 (now renamed ST8sia8) had no SiaNAl transfer activity on the same mammalian substrates. Can the authors provide proof (perhaps just a reference), that fish enzymes would be expected to operate on mammalian substrate? The reciprocal would also reduce my concern – can the authors show that human ST8Sia6 has activity on fish substrates? It seems like the authors acknowledge this in line 275 “These data suggested a loss of function of this enzyme or a low affinity for the used mammalian acceptor substrates...” but is immediately ignored in the next sentence line 276: “Altogether, these data show that the MPSA approach could be used to determine […] activities onto the glycoprotein acceptors” - I don’t believe that statement.
We agree with the reviewer’s comment, we have drawn here a rapid and twofold conclusion that was not clear enough for the reader. In this twofold conclusion, the first point concerned the lower activity (or no activity) of the fish ST8Sia VI-like enzyme towards mammalian substrates and the second point concerns the use of MPSA approach to detect ST8Sias activities.
Concerning the first point: A few fish sialyltransferases enzymes have been cloned and enzymatically tested in in vitro assays using mammalian acceptor substrates. Among these, the Takifugu ST3Gal II (Lehmann et al. 2008 Glyconj J) shows very little transfer activity of [14C]NeuAc on bovine asialofetuin used as an acceptor compared to the rat orthologue used in the same study. Interestingly, the Takifugu ST3Gal II shows no activity towards glycolipids whereas the mammalian counterpart is active on GM1 and GM1b gangliosides suggesting both a lower affinity for the mammalian glycoprotein substrates and an evolution in the enzymatic specificities of this fish orthologue. Similarly, very low transfer activity of [14C]NeuAc onto the mammalian substrate was described for the salmonid polysialyltransferases ST8Sia II and ST8Sia IV (Asahina et al. 2006 J Biochem) and no in vitro autopolysialylation could be detected using these fish enzymes. In addition, the authors reported a cooperative action of the Oncorhynchus ST8Sia II, ST8Sia IV and ST6GalNAc II (Asahina et al 2004 J Biochem) enzymes for the polysialylation of the fish polysialylglycoprotein PSGP.
These few studies favor the idea that the fish sialyltransferases have lower affinity for mammalian glycoprotein acceptors. They suggest also a potential evolution of the enzymatic activity. This idea has been rephrased in the manuscript and references were added.
We actually did not check the reciprocal i.e. lower activity of the human ST8Sia VI enzyme towards fish substrates.
Concerning the second point: We have shown in these experiments that the enzymatic activity of the human ST8Sia VI could be studied with the novel, rapid and efficient multiplate sialyltransferase assay (MPSA) based on the use of various mammalian acceptor substrates and the non-natural CMP-SiaNAl donor substrate. This is the time that this MPSA is extended and adapted to detect activity of an ST8Sia (α2,8-sialyltransferase).
We have reorganized this paragraph and added these sentences (lines 281-300 ) and amended the manuscript accordingly.
Overall comments:
Figure legends: There seems to be an issue with all the figure legends where there are two titles. This is likely an auto-formatting issue from the upload to the journal submission; however some of the repeated titles are not 100% identical - just pointing this out so the editor or authors can pick the correct title. In addition, the figure legends contain too much detail regarding the methods. There is a "MATERIALS AND METHODS" section and this information is also appropriately found there. In addition, there are results and conclusions found in many of the figure legends – which is also more appropriately found in the “RESULTs and DISCUSSION” section and should be removed from the figure legends.
Yes, it was an auto-formatting issue from the upload to the journal submission. All figure legends were revised in the formatted version of the manuscript, with the correct title and shorter caption.
Figure 2. (A) y axis could use a more informative label – “Zebrafish st8sia6-like cDNA (fg)” perhaps? “Q-PCR” is usually written with a lower case “q”.
This modification has been made in figure2A
Figure 4. B is described before A is. Should be switched.
Corrections were made accordingly.
Figure 5. The resolution of my copy of the figure is not high enough for me to read the names – even when viewing the electronic copy. I’m not sure what the coloring scheme means – why are most species names (which I can’t read most of them) blue for the gene families, but ST8Sia V and I are green or purple? If the resolution can’t be improved, then perhaps each species name could be colored and marked in a key or maybe a couple arrows pointing to specific points?
Corrections were made in figure 5, the abbreviated name of animal species were enlarged and are now indicated in black. The color code corresponds to the one described in figure 6 for each subfamily. Color key was added in figure 5.
Table legends: Remove methods from the table legends. This amount of captioning is unusual and is better served in the “MATERIALS AND METHODS” section. Table 1 and Table 2 have the exact same name.
Table1 and table2’s name were changed and caption was amended.
Minor grammatical comments/suggestions:
Line 26: “MPSA sialylation assay” suffers from RAS syndrome (redundant acronym syndrome syndrome). I suggest spelling out the entire acronym in a the abstract, since this is the first time readers will encounter it.
This was corrected accordingly throughout the manuscript.
Line 49: New paragraph beginning with “In teleost fishes…”
This new paragraph was created.
Line 57: “,most of these sialylated structures comprising” is a sentence fragment and needs reworded
A new sentence was made.
Line 81: “substrates” to “substrate”
Sialylmotif regions bind the donor substrate (CMP-Neu5Ac) and the acceptor substrate. So the modification was not brought to the text.
Line 101: this sentence is unclear. It almost reads like snakes and lizards are teleost fishes…
This sentence and the paragraph were made clear in the new version of the manuscript, lines 101-107.
Line 129: add “nucleotide” in front of “sequence” and Line 129: change “seed” to “query”
These 2 changes were made.
Line 129: The BLAST source is very old and is for a protein specific search. PMID: 20003500 is probably more appropriate
We agree with the reviewer that this is an old BLAST source. However, we did not change this reference because it is actually the one BLAST source that was used and is described in the mentioned reference.
Line 131: “exhaustive searches in transcriptomic databases” cites a single database. This does not seem
This change was made
Line 140: “which locations are not conserved neither in…” is awkward.
Corrections were brought to this sentence
Line 171: “quantitatively” is redundant.
The word was deleted
Line 189: Sentence starting with “At larval stage” needs some rewording
A few modifications were made
Line 234: suggest changing “admitted” to something else such as “accepted”
This change was made
Line 258: “We, therefore” - remove the comma
The comma was removed
Line 455: Sources 32 and 35 don’t seem to fit here
In these two references, we described the BLAST search approach that conducted to the ST8Sias identification.
Line 425 and 126 - lowercase vs capitalization difference in headers with "and"
Correction was made line 129
Reviewer 2 Report
In this paper, authors attempted to analyze the function of zebrafish mono-a2,8-sialyltransferase based on human ST8Sia VI. This cloned enzyme was unable to detect the activity by the existing method unlike human ST8Sia VI. The authors showed that human ST8Sia VI orthologue has disappeared in the ray-finned fishes and that the homologue described in fishes correspond to a new subfamily of a2,8-sialyltransferase named ST8Sia VIII using comparative genomics and molecular phylogeny approaches. Nevertheless, I think that the detection of biological activity or function is necessary for the identification of new enzyme subfamilies.
Author Response
Comments and Suggestions for Authors
In this paper, authors attempted to analyze the function of zebrafish mono-a2,8-sialyltransferase based on human ST8Sia VI. This cloned enzyme was unable to detect the activity by the existing method unlike human ST8Sia VI. The authors showed that human ST8Sia VI orthologue has disappeared in the ray-finned fishes and that the homologue described in fishes correspond to a new subfamily of a2,8-sialyltransferase named ST8Sia VIII using comparative genomics and molecular phylogeny approaches. Nevertheless, I think that the detection of biological activity or function is necessary for the identification of new enzyme subfamilies.
We agree with the reviewer that detecting enzymatic activity for this new ST8Sia subfamily would have been ideal. We cloned several isoforms of this fish enzyme and tested its potential activity under several conditions. As discussed in the manuscript, this fish enzyme is probably highly specific of a fish acceptor substrate. Answering this major issue will require developing new experiments using the right substrate and approaches. At least do we show here clearly that it is not the expected orthologue of the human ST8Sia VI enzyme that generate diSia motifs on O-glycans.
Reviewer 3 Report
Review Report
Summary:
Sialic acids are monosaccharides found at the most terminal position in many glycoproteins and glycolipids. They are transferred by sialyl transferases. Chang et al. report the identification and characterization of a novel mono-alpha2,8-sialyltransferase of zebrafish. As shown by RT-PCR and in situ hybridization, ST8SiaVIII is expressed thoughout embryonic development, in the central nervous system as well as in non-neuronal tissues. In vitro enzymatic assays failed to identify an acceptor substrate for this enzyme, so far. Extensive phylogenetic and phylogenomic analysis revealed, that ST8SiaVIII belongs to a newly identified subfamily of the sialyl transferases resulting from the second genome duplication in vertebrates; this subfamily has only been maintained in ray-finned fish.
Broad comments:
In order to understand the regulation and the function of protein and lipid glycosylation, it is of importance to identify and characterize the enzymes involved in the synthesis of glycan structures. To that end, the identification and characterization of a novel sialyl transferase by Chang et al. significantly broadens our knowledge of the mechanisms regulating the sialome, i.e. the total complement of sialic acid types, their linkages and their modes of presentation in a cell
The statistics background of the RT-PCR results is a bit weak. Analyzing only two samples in triplicate does not give valuable statistics. It is also not pointed out whether and how values are normalized. As the RT-PCR results are not discussed in great detail, this does not affect the overall conclusion of the paper.
The enzymatic characterization in 2.3 is sound, but unfortunately fails to identify a substrate acceptor for St8SiaVIII. This is interpreted as “loss of function” in line 275 and as “the nature of acceptor substrate …. still awaits identification” in line 286. As the correct acceptor might not have be contained in the tested collection of possible substrate, the first interpretation seems a bit courageous and the latter interpretation should be favoured.
Overall: the identification and in-depth sequence analysis as well as the impressive phylogenetic characterization of a new zebrafish sialyltransferase and its vertebrate ortholog add important knowledge to the understanding the biology of sialic acids in animal development. Yet, the functional characterization of the novel zebrafish St8SiaVIII needs to be deepened in future work.
The mamuscript has a high quality, only minor revisions are necessary as pointed out above and below.
Secific Comments:
Lines 150, 160 169, 174 and many more – the authors isolated and identified a gene, initially found by its similarity to St8Sia VI. Concluding from the phylogenetic analysis , it is renamed ST8SiaVIII from line 347 on. The nomenclature in previous parts of the manuscript is not consistent (“St8SiaVI-like” in many cases, “St8SiaVI” in line 160, “St8Sia” in line 168 and 169, ….). This should be made consistent.
Also: in Fig. 5, the group of ST8-SiaVI-like genes is named “Fish ST8SiaVIII “, although this new nomenclature has not been introduced in the text up to that point. In Fig.S1, the ST8SiaVIII group contains genes named St8Sia8, in Fig. S2 genes in this group are named St8Sia6. In Fig.5, sequence names are illegible, even if you zoom in on the PDV version provided. This should be made consistent.
Line 414: the “disrupted line” framing the st8sia7 box in Fig. 7 is barely visible
Author Response
Summary:
Sialic acids are monosaccharides found at the most terminal position in many glycoproteins and glycolipids. They are transferred by sialyl transferases. Chang et al. report the identification and characterization of a novel mono-alpha2,8-sialyltransferase of zebrafish. As shown by RT-PCR and in situ hybridization, ST8SiaVIII is expressed thoughout embryonic development, in the central nervous system as well as in non-neuronal tissues. In vitro enzymatic assays failed to identify an acceptor substrate for this enzyme, so far. Extensive phylogenetic and phylogenomic analysis revealed, that ST8SiaVIII belongs to a newly identified subfamily of the sialyl transferases resulting from the second genome duplication in vertebrates; this subfamily has only been maintained in ray-finned fish.
Broad comments:
In order to understand the regulation and the function of protein and lipid glycosylation, it is of importance to identify and characterize the enzymes involved in the synthesis of glycan structures. To that end, the identification and characterization of a novel sialyl transferase by Chang et al. significantly broadens our knowledge of the mechanisms regulating the sialome, i.e. the total complement of sialic acid types, their linkages and their modes of presentation in a cell
The statistics background of the RT-PCR results is a bit weak. Analyzing only two samples in triplicate does not give valuable statistics. It is also not pointed out whether and how values are normalized. As the RT-PCR results are not discussed in great detail, this does not affect the overall conclusion of the paper.
We wanted to make sure that this gene was expressed during embryogenesis prior whole mount in situ hybridization analysis. We did absolute quantification using a plasmid containing the sequence of interest.
The enzymatic characterization in 2.3 is sound, but unfortunately fails to identify a substrate acceptor for St8SiaVIII. This is interpreted as “loss of function” in line 275 and as “the nature of acceptor substrate …. still awaits identification” in line 286. As the correct acceptor might not have be contained in the tested collection of possible substrate, the first interpretation seems a bit courageous and the latter interpretation should be favoured.
We totally agree with the reviewer. The paragraph was reorganized and this discussion was made clearer in the manuscript.
Overall: the identification and in-depth sequence analysis as well as the impressive phylogenetic characterization of a new zebrafish sialyltransferase and its vertebrate ortholog add important knowledge to the understanding the biology of sialic acids in animal development. Yet, the functional characterization of the novel zebrafish St8SiaVIII needs to be deepened in future work.
The mamuscript has a high quality, only minor revisions are necessary as pointed out above and below.
Secific Comments:
Lines 150, 160 169, 174 and many more – the authors isolated and identified a gene, initially found by its similarity to St8Sia VI. Concluding from the phylogenetic analysis , it is renamed ST8SiaVIII from line 347 on. The nomenclature in previous parts of the manuscript is not consistent (“St8SiaVI-like” in many cases, “St8SiaVI” in line 160, “St8Sia” in line 168 and 169, ….). This should be made consistent.
We have made the requested changes: the fish enzyme is named ST8Sia VI-like before line 347, and after the phylogenetic characterization it is renamed ST8Sia VIII.
Also: in Fig. 5, the group of ST8-SiaVI-like genes is named “Fish ST8SiaVIII “, although this new nomenclature has not been introduced in the text up to that point. In Fig.S1, the ST8SiaVIII group contains genes named St8Sia8, in Fig. S2 genes in this group are named St8Sia6. In Fig.5, sequence names are illegible, even if you zoom in on the PDV version provided. This should be made consistent.
The name of the new subfamily was changed to ST8Sia VI-like in figure 5 and in supplemental figure S1 and S2. The color code was homogenized and a color key was added. Sequence names were enlarged and made visible.
Line 414: the “disrupted line” framing the st8sia7 box in Fig. 7 is barely visible
Figure 7 was redrawn and the disrupted line is now visible.
Round 2
Reviewer 2 Report
The authors have not responded to my request, but I understood the authors’ opinions. I agree that this paper be accepted.